# Nanomaterial-Based Fluorescent Biosensor for Food Safety Analysis

**DOI:** 10.3390/bios12121072

**Published:** 2022-11-23

**Authors:** Jiaojiao Zhou, Yue Gui, Xuqin Lv, Jiangling He, Fang Xie, Jinjie Li, Jie Cai

**Affiliations:** 1National R&D Center for Se-Rich Agricultural Products Processing, Hubei Engineering Research Center for Deep Processing of Green Se-Rich Agricultural Products, School of Modern Industry for Selenium Science and Engineering, Wuhan Polytechnic University, Wuhan 430023, China; 2Key Laboratory for Deep Processing of Major Grain and Oil, Ministry of Education, Hubei Key Laboratory for Processing and Transformation of Agricultural Products, Wuhan Polytechnic University, Wuhan 430023, China; 3Institute of System and Engineering, Beijing 100010, China

**Keywords:** fluorescence, food safety analysis, biosensors, nanomaterials

## Abstract

Food safety issues have become a major threat to public health and have garnered considerable attention. Rapid and effective detection methods are crucial for ensuring food safety. Recently, nanostructured fluorescent materials have shown considerable potential for monitoring the quality and safety of food because of their fascinating optical characteristics at the nanoscale. In this review, we first introduce biomaterials and nanomaterials for food safety analysis. Subsequently, we perform a comprehensive analysis of food safety using fluorescent biosensors based on nanomaterials, including mycotoxins, heavy metals, antibiotics, pesticide residues, foodborne pathogens, and illegal additives. Finally, we provide new insights and discuss future approaches for the development of food safety detection, with the aim of improving fluorescence detection methods for the practical application of nanomaterials to ensure food safety and protect human health.

## 1. Introduction

Food safety is a worldwide concern and has attracted considerable attention. Currently, there are many laws and regulations that ensure food safety. For example, the International Organization for Standardization (ISO) has published standards for the food industry [1]. Although relevant measures have been taken to guarantee food safety, food safety issues frequently threaten human health (Figure 1). For instance, the insecticide fipronil was found in eggs produced in Belgium and the Netherlands [2]. This food safety problem has gathered considerable attention in western Europe. Recently, Wang et al. reported clenbuterol food poisoning in 13 patients who consumed snake meat. They suffered from headaches, palpitations, tachycardia, hypokalemia, and other symptoms within the first 3 h after exposure [3]. These incidents and *Salmonella* outbreaks demonstrate the importance of food safety analysis. Facing such issues, it is necessary to guarantee food safety and reduce risks. Therefore, food safety analysis is critical. 

A literature survey using the Web of Science shows that more than 100,000 journal articles have been published about food safety after 2011. In recent decades, various methods have been developed for food safety monitoring, including gas chromatography [4], liquid chromatography [5], mass spectrometry [6], and gas chromatography–mass spectrometry [7]. Although the aforementioned traditional methods can achieve satisfactory results, they require expensive instruments, professional operators, and tedious pretreatment steps, limiting their practical applications [8]. Therefore, developing rapid, simple, and efficient methods for food safety analysis is imperative.

A biosensor is an analytical strategy that can be employed to convert an interaction into a physically detectable signal and selectively and sensitively detect a target [9]. Biosensors are ideal candidates for food safety analysis owing to their rapid detection, simplicity, low cost, and portability. Various biosensors have been developed to satisfy food inspection requirements [10].

Common biosensors for food safety analysis include electrochemical, fluorescent, surface-enhanced Raman spectrometry, and colorimetric biosensors. A few others (surface plasmon resonance (SPR) spectroscopy, thermometric, and piezoelectric biosensors) have also been reported. Some of these can detect targets in complex food matrices and satisfy the requirements of trace-level detection. In particular, fluorescent biosensors have attracted considerable attention owing to their simplicity, rapidity, accuracy, and in situ detection [11]. Furthermore, their spectral properties and fluorescence anisotropy can be used alone or together for the signal output [12]. These advantages make them suitable for use in food safety analysis.

Advances in nanotechnology and materials science have yielded numerous novel nanomaterials that provide potential opportunities for biosensing. Nanomaterials can significantly improve the performance of biosensors, including their sensitivity, selectivity, and accuracy, which can facilitate the development of biosensing for food safety detection [13].

Several reviews have been published on food safety detection [14,15,16,17,18]; however, food safety detection using fluorescent biosensors based on nanomaterials is scarcely reviewed in detail. Herein, we first describe biomaterials and nanomaterials used in fluorescent biosensors (Figure 2). We then comprehensively summarize recent developments in nanomaterial-based fluorescent biosensors for food safety analysis, including mycotoxins, heavy metals, antibiotics, pesticide residues, foodborne pathogens, and other illegal additives. Finally, the challenges and future perspectives of food safety analysis are discussed. This study provides guidance for research on nanomaterial-based fluorescent biosensors for food safety analysis. We expect that more nanomaterials and biosensing strategies will be introduced, which may stimulate breakthroughs in this field.

## 2. Biomaterials Used in Food Safety Analysis

### 2.1. Aptamers

Aptamers are single-stranded nucleic acids that selectively bind to target molecules [19]. In contrast to antibodies, aptamers can be chemically synthesized, and their production does not rely on biological systems. Thus, they have high batch-to-batch reproducibility and can easily be labeled at arbitrary positions, e.g., by modification with a thiol or carboxyl group. Compared to antibodies, aptamers have low immunogenicity and toxicity. Numerous studies have been conducted on aptamer selection and related biosensing applications [20]. We are optimistic that aptamer-based biosensors will continue to be developed because of the wide selection of aptamers, advances in material science, and nanotechnology. Recently, studies on the use of aptamers for cancer therapy have been performed [21,22,23]. We believe it is time to progress to the next aptamer-based biosensing and biomedicine stage.

Numerous aptasensors have been proposed for food safety monitoring, including aptamer-based colorimetric [24], fluorescent [12,25], electrochemical [26,27], and SPR biosensors [28]. Wang et al. used an aptamer screening kit to rapidly screen aptamers of foodborne Vibrio parahaemolyticus [29]. Additionally, the aptamer screening kit offers a significantly shorter screening time. The selected aptamers can be combined with gold nanoparticles (AuNPs) to produce more intuitive results. These methods and techniques open new avenues for the efficient and rapid determination of bacteria. Another study was based on the inherent capability of an aptamer to hybridize to a complementary strand in addition to binding to a target.

In conclusion, aptamers have received increasing attention for affinity-based assays owing to their high affinity, specificity, stability, and ease of chemical synthesis [30]. They have great potential for application in the field of food safety.

### 2.2. Antibodies

With the exception of aptamers, antibodies are the most commonly used recognition molecules because they can selectively bind to analytes with a high affinity [31].

The production of antibodies involves animals. In addition, antibodies have high molecular weights. Immunoassays are commonly used in biosensing. Antibody-based assays require immobilization and washing steps. Antibody-based commercial assay kits are widely used to analyze small molecules, viruses, and cancer biomarkers. The preparation of high-quality antibodies is key for the development of immunoassays [32]. Furthermore, multi-analyte antibodies are favored because they simultaneously determine several targets [33]. Sheng et al. reported a broad-spectrum heterocyclic aromatic amine (HAA) antibody that could be used to detect HAA residues in heated meat [34].

With advancements in biology and immunology, the performance of antibodies has constantly improved, and they are expected to play an important role in food safety analysis.

### 2.3. Enzymes

Enzymes are excellent biomolecules for biosensor development. In contrast to biorecognition elements, enzymes are not used to separate or immobilize analytes but to catalyze reactions for signal generation and amplification [2]. Horseradish peroxidase (HRP) is a representative enzyme used in food analysis. Enzymes are used to detect metal ions [35], small molecules [36], pesticides [37], mycotoxins [38], and drugs [39].

Enzymes are not negatively affected by the operating environment, making them preferred candidates for manufacturing efficient biosensors [40]. The most successful application of enzymes in biosensors is glucose meters, in which glucose oxidase catalyzes glucose to produce gluconic acid and H_2_O_2_. However, these enzymes suffer from low stability and high cost. In 2007, the first study on nanozymes was performed by Yan et al. [41]. Since then, nanozymes have attracted considerable research attention.

Nanozymes, such as carbon, metals, and their oxides, as well as two-dimensional materials, can be used for real-time food safety detection and monitoring. For example, Zhang et al. prepared an AuNP-labeled aptamer (Au-apt) for ampicillin (AMP) detection [42]. Au-apt was synthesized as a labeled antibody for colorimetric catalysis, and AuNPs in the stationary phase were used to catalyze color development. The degree of color development was inversely proportional to the AMP content. This method is based on a nanozyme-linked immunosorbent assay with simple material preparation and high selectivity and sensitivity.

### 2.4. Nucleic Acids

Nucleic acids are natural biopolymers of nucleotides [43]. With the rapid development of biotechnology and DNA nanotechnology, nucleic acids have a wide range of applications in biosensing, bioimaging, and biomedicine. Xia et al. found that isothermal nucleic-acid amplification is useful for the rapid and efficient detection of targets [44]. Chen et al. developed a colorimetric method for protein determination via the assembly of nucleic acids and proteins [45]. Nucleic-acid assembly was triggered by biotinylated DNA chains after sandwich immunoreactions. This colorimetric sensor enabled the quantitative determination of proteins. Although the application of nucleic acid-based sensors for food safety analysis is just beginning, they have considerable application potential because of their specific recognition ability and high sensitivity.

Nucleic acids with catalytic activities similar to enzymes are called DNAzymes [46]. Lee et al. immobilized uranyl ion (UO_2_^2+^)-specific DNAzymes onto AuNPs. The DNAzyme complex was formed by hybridization between the enzyme strand and fluorescent-labeled substrates. In the presence of UO_2_^2+^, the fluorescent substrates were released from the AuNPs, leading to increased fluorescence. Thus, UO_2_^2+^ was detected in cancer cells [47].

### 2.5. Others

The single-stranded DNA binding protein uses a unique folding pattern to specifically bind to single-stranded DNA. Miao reported a fluorescent biosensor for chloramphenicol (CAP) assay based on a magnetic aptamer-liposome vesicle probe [48] (Figure 3). Specifically, a single-stranded DNA (ssDNA)-binding protein bound to the corresponding complementary DNA. Aptamer-labeled magnetic beads (MBs) were used as capture probes, hybridizing with ssDNA-binding proteins on the liposome surface. When CAP is present, the aptamer is released from the liposomes. A fluorescence signal corresponding to the concentration of CAP was observed in the supernatant after magnetic separation.

## 3. Nanomaterials in Food Safety Analysis

### 3.1. Nanoparticles

Nanoparticles typically range in size from 1 to 100 nm, and they exhibit size-dependent properties [49] and have attracted increasing attention. Most metal nanoparticles, such as gold, silver, and platinum nanoparticles, have useful effects, such as electron conductivity or plasmon production. They are frequently used as labels in colorimetric, fluorescent, surface-enhanced Raman spectroscopy, electrochemical, electrochemiluminescent, and photoelectrochemical biosensors. Additionally, nonmetallic nanoparticles, such as carbon dots (CDs), silica nanoparticles, and polymer nanoparticles, are widely used in various biosensors.

The most commonly used materials are AuNPs, which detect small molecules, proteins, nucleic acids, viruses, and bacteria. AuNPs are attractive for their plasmonic effects depending on their size and are therefore used as labels in various biosensors [50]. In addition to their optical properties, their enzyme-mimicking activities, such as peroxidase-like activity, have drawn much attention for immunoassays and other analytical applications because they can replace traditional HRP in the enzyme-linked immunosorbent assay (ELISA) [51,52]. Many biosensors for glucose and H_2_O_2_ detection using the glucose oxidase-like activity of AuNPs to convert glucose into gluconic acid and H_2_O_2_ have been proposed [53,54,55]. Additionally, Zhang et al. observed the dehydrogenase-mimicking activity of AuNPs when estradiol was used as a substrate [56].

Similarly, silver, platinum, and bimetallic alloy nanoparticles have been employed in biosensing. Chen et al. used DNA to regulate the growth of platinum nanoparticles (PtNPs) on graphene oxide (GO-PtNPs) [57]. A hybridization chain reaction (HCR) was triggered in the presence of the target, leading to the formation of a double-stranded DNA (dsDNA) structure, which hindered the adsorption of free ssDNA on the surface of GO, thereby accelerating the growth of PtNPs. The formed GO-PtNPs exhibited excellent catalytic activity toward chromogenic substrates, generating a colorimetric signal. In addition, because of their high electron conductivity, nanoparticles are often used as electrode materials in electrochemical, electrochemiluminescent, and photoelectrochemical biosensors. Magnetic nanoparticles have been used in biosensing because of their good electron conductivity and magnetism [58,59].

Luminescent nanoparticles, such as quantum dots (QDs), upconversion nanoparticles (UCNPs), and carbon nanomaterials, can also be employed for signal generation. They are primarily used as fluorescent labels. Zheng et al. used functional magnetic QDs to capture immune probes in a multiplexed immunochromatographic assay [60]. Owing to the signal amplification of multilayer QD shells and the enrichment of targets, this method could effectively capture three target mycotoxins from complex systems. In addition, these materials are utilized as electrode materials owing to their excellent conductivities.

### 3.2. Nanoclusters

Sub-nanometer metal clusters have attracted considerable attention in food safety monitoring because of their favorable optical, electronic, and catalytic properties. Gold nanoclusters (AuNCs) are the most widely used because of their ultra-small size, tunable emission, size-dependent fluorescence, and good biocompatibility [61]. Using disulfide-bond-functionalized AuNCs (S-S-AuNCs) as the emitter, Li et al. proposed a new fluorescence biosensor for methidathion detection [62]. S-S-AuNCs were synthesized via the redox reaction of dithiothreitol with HAuCl_4_. They exhibited a good response to mercaptan compounds and a low pH value. This study provided a simpler method for organophosphorus pesticide (OP) assays. Silver nanoclusters also have considerable application prospects as substitutes for traditional probes and sensors. Obliosca et al. reported that silver nanoclusters (AgNCs) could be used as fluorescent nanomaterials for the sensitive and specific detection of DNA [63]. Copper nanoclusters (CuNCs) are also used in food monitoring, and Bagheri et al. proposed a dual detection sensor using aptamer- and antibiotic-based methods combined with CuNCs [64] as an effective method for the identification and quantification of *Staphylococcus aureus* (*S. aureus*) pathogens. When *S. aureus* was used as a specific target to induce the aggregation of CuNC-labeled aptamer, the interaction between them limited the internal movement of molecules and provided energy for luminescence. *S. aureus* can be detected using fluorescence enhancement. In addition to individual nanoclusters (NCs), bimetallic NCs have been used, which combine the advantages of the two materials.

Compared to their larger counterparts, nanoclusters have attractive biological, optical, and chemical properties.

### 3.3. Two-Dimensional Materials

Two-dimensional (2D) materials represent another group of synthetic materials that have received much interest owing to their tunable electronic, optical, ferromagnetic, and chemical properties [65]. The family of 2D materials is growing rapidly and now includes GO and transition-metal dichalcogenides as well as chemical derivatives, hybrids, and allotropes [66].

The predominant 2D material is GO. GO, and carbon nanotubes can be employed as electrode materials to enhance the electrode surface, serve as carriers for signal amplification, or provide specific binding to the target. Ma et al. used a graphene nanosheet to immobilize abundant proteins for signal amplification owing to its large specific surface area [67]. Shi et al. modified MXenes with DNA to control the growth of PtNPs for the colorimetric sensing of Hg^2+^ [68] (Figure 4). The prepared MXene/DNA/Pt exhibited peroxidase-mimicking activity in a sequence-dependent manner. Hg^2+^ can inhibit this peroxidase-like activity because of the partial reduction of Hg^2+^ to Hg^0^. Coupled with a microfluidic system, this sensor achieved the point-of-care detection of Hg^2+^.

Other 2D nanomaterials, such as manganese dioxide (MnO_2_) nanosheets, have been widely used in biosensing applications. MnO_2_ nanosheets are generally used as fluorescence quenchers in fluorescent biosensors. Yuan et al. used MnO_2_ nanosheets as label-free nanoplatforms to simultaneously detect Ochratoxin A (OTA) and cathepsin D [69]. This biosensor exhibited excellent sensing performance and robustness in complex sample matrices due to the strong optical absorption and fast electron transfer capabilities of MnO_2_ nanosheets. Similar to MnO_2_ nanosheets, tungsten disulfide (WS_2_) nanosheets have been used as fluorescence quenchers for the detection of microRNAs [70].

The development and application of 2D nanomaterial-based sensors provide insight into heavy-metal ion and small-molecule detection. There is room for further development in food safety analysis.

### 3.4. Metal-Organic Frameworks

Metal-organic frameworks (MOFs) are constructed from metal ions and organic linkers [71]. These materials have large surface areas, high porosities, and tunable sizes, showing promise for catalysis, gas storage, gas separation, chemical sensing, and drug delivery [72]. As an emerging class of adsorbents, MOFs can adsorb contaminants from food and the environment. Shirin et al. found that MOFs exhibit binary functions. They can be used as rapid colorimetric and fluorescence probes and as effective adsorbents for mercury ions. With the adsorption of Hg^2+^, the color of the sorbent changes from light cream to fluorescent yellow [73].

In addition to the adsorption of contaminants, MOFs have been used in various biosensors. For example, Zhang et al. synthesized a ZrMOF with a large specific surface area and applied it to an enzyme-free electrochemical biosensor for *Pseudomonas aeruginosa* detection [74]. A Cu-ZrMOF catalyst with high catalytic activity was synthesized by adding a certain amount of Cu^2+^. A Cu-ZrMOF@Aptamer@DNA nanocomposite was then synthesized by catalyzing the decomposition of H_2_O_2_ and used as a signal probe. Upon the addition of *Pseudomonas aeruginosa*, an enhanced current signal was detected.

MOFs are the fastest-growing class of materials in chemistry and have been widely employed in food quality control, storage, and shelf-life management. In the future, this technology will continue to flourish.

### 3.5. Others

Many other materials have also been used for food safety analysis, including biopolymers, dendrimers, fullerenes, molecularly imprinted polymers (MIPs), and nanocontainers. These materials are used to construct alternative sensor platforms for the sensitive and reliable determination of food contaminants.

MIPs can be used for the separation of analytes from complex matrices and binding to the target owing to their specific recognition. By incorporating the donor and acceptor into a silica nanosphere, Chen et al. developed a solid-state MIP resonance energy transfer (RET) sensor for the sensitive detection of α-ergocryptine and OTA [75] (Figure 5). When contaminants are present, the MIP cavities rebind to the analytes, inducing a quenched electrochemiluminescence (ECL) signal. Despite their high mechanical strength and robustness, MIPs may suffer from imprint leakage, which reduces their sensitivity [2]. MIPs have been used to detect small molecules [76,77]. Additionally, MIPs coupled with magnetic materials can achieve integrated separation and detection. Li et al. imprinted a polymer shell onto magnetic nanoparticles [78]. Due to the target rebinding and magnetic separation, these polymers can detect triazophos in vegetable samples.

Similar to MOFs that serve as hosts, nanocontainers, such as liposomes and core–shell structured nanoparticles, are used in fluorescence biosensors. Magnetic mesoporous particles are composed of silica spheres. Zhu et al. immobilized acetylcholinesterase (AChE) on the surface of Fe_3_O_4_@SiO_2_@mSiO_2_ via covalent bonds [79]. They act as fluorescence tracers for carbamate pesticide detection in Chinese cabbage and cucumber plants.

In addition to these materials, numerous other composites have been used for food safety analysis. For instance, in combination with Au-Ag Janus nanoparticles and Au nanobipyramids, CsPbBr_3_@mesoporous silica nanomaterials and ZnGeGaO:Cr,Er,Yb nanoparticles have been separately used as carriers of antibodies for signal amplification in immunoassays for *S. aureus* enterotoxin detection [80,81]. Similarly, nanozymes have been successfully applied in food safety assays because of their high catalytic activities [82,83].

## 4. Application of Fluorescent Biosensors for Food Safety Analysis

### 4.1. Mycotoxins

Mycotoxins are secondary metabolites produced by toxigenic fungi [84]. They accumulate in organisms throughout the food chain. Owing to their teratogenic, mutagenic, immunosuppressive, and carcinogenic properties [8], they may cause damage to organisms. Therefore, mycotoxin contamination has attracted considerable attention.

T-2 toxin is one of the most toxic type A trichothecene mycotoxins produced by several Fusarium species [85]. Khan et al. fabricated a fluorescent aptasensor to analyze the T-2 toxin [86]. A fluorescence resonance energy transfer (FRET) aptasensor was proposed for detecting T-2 toxin using aptamer-modified AgNCs as the donor and molybdenum disulfide (MoS_2_) nanotablets as the acceptor. The presence of T-2 toxin weakens the interaction between AgNCs and MoS_2_, preventing FRET and inducing the gradual recovery of the fluorescence signal. Compared to traditional biosensors, aptasensors have the advantages of simple preparation, high sensitivity, and good selectivity. In another work, on the basis of FRET between QDs and AuNPs, Goryacheva et al. proposed a homogenous FRET-based fluorescent immunoassay for deoxynivalenol (DON) detection [87]. Fumonisins are a group of mycotoxins produced by *Fusarium* species that are found worldwide, mainly in maize [88]. Peltomaa et al. proposed a homogeneous fluorescence immunoassay based on the fluorescence quenching ability of AuNPs and a recombinant epitope-mimicking fusion protein for the detection of mycotoxin fumonisin B_1_ (FB_1_) [89]. The fumonisin mimotope was used as a fluorescent protein for FB_1_ detection without the need for a secondary antibody. This immunoassay could be carried out in a single step without cumbersome washing steps. There was no significant cross-reactivity with other mycotoxins, and acceptable recoveries were obtained from spiked wheat samples. This indicates a great promise for the simple analysis of mycotoxin-contaminated food samples.

Multiple mycotoxins may coexist in the food matrix, and their detection has been reported. Yan et al. proposed a QD/QD-microbead-based multiplex immunochromatographic assay for the simultaneous detection of FB_1_, zearalenone (ZEN), and OTA [90] (Figure 6). QDs and QD microbeads were chosen as the detection probe after modification with antibodies. The detection limits were 0.25 ng/mL for FB_1_, 3.0 ng/mL for ZEN, and 0.5 ng/mL for OTA, respectively. This immunosensor displayed good accuracy and repeatability for practical sample sensing. Hence, this method has the potential for the quantitative multiplex detection of mycotoxins.

In addition to QDs, TiO_2_ and hydrogels have been used to detect multiple mycotoxins. Ji et al. reported a real-time mycotoxin detection system based on a shape-encoded hydrogel and portable smartphone devices [91]. This hydrogel system has the advantage of preparing high-pass quantitation. A simple assembled point-of-care test device can obtain the test results using a smartphone. Liu et al. proposed an aptamer microarray to simultaneously screen multiplex mycotoxins on a TiO_2_−porous silicon surface via hybridization between the aptamer and complementary DNA [92]. The deposition of a TiO_2_ nanolayer on the surface of the porous silicon resulted in a 14-fold enhanced fluorescence intensity relative to that of the individual porous silicon. The microarray achieved the simultaneous screening of multiplex mycotoxins with detection limits of 15.4 pg/mL for OTA, 1.48 pg/mL for AFB1, and 0.21 pg/mL for FB1, respectively.

### 4.2. Heavy Metals

Heavy-metal ions, such as Hg^2+^, Pb^2+^, and Cd^2+^, have attracted considerable attention owing to their significant threat to public health. In addition, the environment can be severely polluted if heavy metals are directly piped into rivers or soil, eventually entering the human body [93]. Therefore, developing reliable and simple methods for the detection of trace heavy metals is important for environmental research, food research, agriculture, industry, and other fields.

Mercury is a highly toxic contaminant that accumulates in organisms via the food chain, causing irreversible damage to humans and ecosystems, even at low concentrations [94]. Since Hg^2+^ stabilizes the thymine-thymine base pair, this interaction enables the development of Hg^2+^-specific biosensors. Pi et al. designed a Hg^2+^ sensor based on the selective binding of Hg^2+^ to a thymine-rich DNA [95]. The interactions between the DNA and SYBR Green I produces the detected fluorescence signal. The results illustrate the feasibility of applying the DNA-based Hg^2+^ sensor technology in freshwater environments. Wang et al. prepared multifunctional inorganic–organic nanocomposites by encapsulating CdTe QDs and rhodamine 6G into magnetic silica nanocomposites [94]. The nanocomposites achieved regenerative ratiometric fluorescence sensing of Hg^2+^. There was a good linear range from 7 to 900 nM for the Hg^2+^ assay. Based on magnetic mesoporous silica nanocomposites, this strategy has high sensitivity, selectivity, and good reusability.

Pb ions (Pb^2+^) are highly toxic heavy metals that can cause serious damage to human health. Based on the reversible fluorescence switching of MOF (NH2-MIL-125(Ti)), Venkateswarlu et al. proposed a “turn-on/off” fluorescent biosensor for Pb^2+^ sensing [96]. In the presence of Pb^2+^, the fluorescence of NH_2_-MIL-125(Ti) was quenched because of the binding of Pb^2+^ to it. Thus, a fluorescent biosensor for Pb^2+^ assays was proposed. Bain et al. reported a green synthesis of AuNCs with tunable emission wavelength by core etching and ligand exchange method for Pb^2+^ assay [97]. The highly green-emitting AuNCs are more sensitive than orange-emitting AuNCs toward Pb^2+^, leading to improved detection sensitivity. The prepared AuNCs nanoprobe exhibited a specific fluorescence response to Pb^2+^ with a detection limit of 10 nM. Yang et al. constructed a fluorescent platform based on zeolitic imidazolate framework-8 (ZIF-8) and a DNAzyme for Pb^2+^ analysis [98]. The fluorescence of the Pb^2+^-dependent DNAzyme was quenched by ZIF-8 owing to the adsorption of the DNAzyme onto ZIF-8. Pb^2+^ can activate and cleave the DNAzyme, releasing 6-carboxyfluorescein (FAM)-labeled DNA. As a result, the fluorescence of the system increased in the range of 0.01–10.0 nM for the Pb^2+^ assay.

Cd(II) is a heavy-metal contaminant that can have chronic effects on humans. To increase the sensitivity of Cd^2+^ detection, Xu et al. used an HCR-assisted DNA amplification strategy involving a Cd^2+^-specific DNA aptamer for Cd^2+^ sensing [99]. The Cd^2+^ aptamer (S0) was used to recognize Cd^2+^ and trigger the HCR. In the absence of Cd^2+^, S0 initiates the HCR, leading to the formation of a dsDNA structure accompanied by a quenched fluorescence signal. Cd^2+^ combines with S0 to block HCR and restore fluorescence. The linear range was 0–10 nM. Zeng et al. synthesized V_6_O_13_ nanobelts with an excellent peroxidase-mimicking activity using a hydrothermal method [100]. A label-free biosensor was fabricated for Cd(II) detection. Heavy metals, such as Cd(II), can inhibit the nanozyme activity of V_6_O_13_ nanobelts, leading to a weakened colorimetric signal. The limit of detection (LOD) of the Cd(II) assay was 1.89 μg/L.

There have been reports of the determination of multiple heavy metals. Lu et al. prepared ion-imprinted fluorescent polymers using blue and red dual-emission CDs [101]. Because Cr^3+^ and Pb^2+^ can separately quench their fluorescence, the simultaneous detection of Cr^3+^ and Pb^2+^ can be achieved using the as-prepared imprinted fluorescence polymers (Figure 7). This biosensor achieved detection limits of 27 and 34 nM for Cr^3+^ and Pb^2+^, respectively. Liu et al. proposed a sensor array to detect Cd^2+^, Hg^2+^, and Pb^2+^ by modifying phosphorothioate with a lanthanide-dependent DNAzyme (Ce13d) [102]. The well-designed DNAzymes exhibited detection limits of 4.8, 2.0, and 0.1 nM for Cd^2+^, Hg^2+^, and Pb^2+^, respectively. Using metal–polydopamine (MPDA) frameworks as fluorescence quenchers, a fluorescence biosensor was established for the selective recognition of Hg^2+^ and Ag^+^ [103]. FAM-labeled ssDNA was adsorbed onto MPDA via its phosphate backbone. After this, the fluorescence of ssDNA was quenched, leading to a significantly weakened fluorescence signal. The addition of Hg^2+^ induced the release of the ssDNA from MPDA owing to the specific binding between Hg^2+^ and MPDA, resulting in an amplified signal. Exonuclease III was used for target recycling to achieve signal amplification. Under optimal conditions, the detection limits of Hg^2+^ and Ag^+^ were as low as 1.3 and 34 pM, respectively.

### 4.3. Antibiotics

Antibiotics are used not only in animal and poultry farming but also in the treatment of human diseases. The abuse of antibiotics leads to a series of health and environmental problems owing to their serious side effects [104,105]. Adverse human reactions to antibiotics, such as allergies, organ damage, and bacterial drug resistance, have raised popular concerns [106]. Therefore, it is necessary to accurately monitor the trace amounts of antibiotics in food products.

Tetracycline (TET) is widely used to inhibit bacterial growth. However, it may enter the human body through the food chain, which threatens public health [107]. Hong et al. fabricated TET-aptamer-pendant DNA tetrahedral nanostructure-functionalized MBs (Apt-tet MBs) as detection probes for TET assays [108]. The introduction of TET triggered the release of the primer from the Apt-tet MBs owing to the specific binding between TET and the aptamer. The released primer then initiated a rolling circle amplification (RCA) reaction, followed by the formation of a long sequence. Using SYBR Green I as the fluorescent material, the fluorescence intensity was monitored via hybridization between the detection probe and RCA product. A linear range of 0.001–10 ng/mL was obtained for the TET assay. Yang et al. reported a ratiometric fluorescence sensor for TET using boron nitride QDs and europium ions (BNQD-Eu^3+^) as fluorescence emitters [109]. The fluorescence of the BNQDs was quenched, and that of Eu^3+^ was enhanced in the presence of TET. The detection limits for TET, oxytetracycline (OTC), and doxycycline (DOX) were 0.019, 0.104, and 0.028 μM, respectively. Mousavizadegan et al. reported a fluorescent biosensor for TET detection based on bovine serum albumin (BSA)-protected Au/Ag bimetallic nanoclusters (BSA-BMNCs) [110]. The interaction of TET with BSA changed the emission of the BSA-BMNCs within the increasing concentrations of TET. Images of the sensing platform were then captured with various smartphones.

Aminoglycoside antibiotics can be used to treat various bacterial infections, including kanamycin (KAN), streptomycin, gentamicin, tobramycin, and neomycin [111]. Zhang et al. proposed a multivalent aptamer probe for kanamycin sensing [112]. The 2-valence aptamer probes provided multiple binding sites, achieving a LOD of 0.097 nM for kanamycin. Wang et al. constructed an aptasensor for tobramycin based on the synergy between HCR and fluorescence. Tobramycin preferentially binds to aptamers on MBs, causing the release of FAM-labeled complementary DNA (cDNA-FAM) [113]. After magnetic separation, the released cDNA-FAM triggers HCR amplification, followed by the formation of long dsDNA. Fluorescence was increased by the insertion of SYBR Green I into the dsDNA. This aptasensor could measure tobramycin concentrations in the range of 0.3–50 μM.

β-lactam antibiotics, such as AMP and penicillin, are extensively used to treat bacterial infections because they inhibit bacterial cell wall synthesis [114]. These residues can enter the human body and damage human health. Chen et al. reported a FRET aptasensor using UCNPs as donors and AuNPs as acceptors in an AMP assay [115]. A good linear range was obtained from 10 to 100 ng/mL for AMP. Jalili et al. designed an eco-friendly visual ratiometric fluorescent sensor for penicillin based on various colored CDs and MIPs [116]. Upon the addition of penicillin, only the fluorescence of the yellow-emissive CDs was quenched owing to target hindrance, whereas that of the blue-emissive CDs remained stable, which resulted in an obvious fluorescence change. A LOD of 0.34 nM was achieved with a 1–32 nM linear response range. CAP is widely used to treat bacterial infections [117]. However, CAP abuse is associated with side effects in humans. Therefore, CAP use has been prohibited in many countries and regions [118]. Chen et al. reported a fluorescence aptasensor that uses DNA-assisted signal amplification for CAP detection. The presence of CAP induces dissociation between the aptamer and corresponding complementary DNA [119]. This leads to the formation of four-arm DNA junctions. After treatment with T7 exonuclease and SYBR Green I, an increased fluorescence intensity was observed.

To simultaneously detect multiple antibiotics, Mandal et al. proposed a carbon nanoparticle-based nine-channel fluorescence array integrated with artificial intelligence for the detection of different class antibiotics [120]. The fluorescence responses of the arrays to six antibiotics (β-lactams: AMP, Quinolones: Ciprofloxacin (CPFX), Aminoglycosides: KAN, Sulfonamides: Sulphamethoxazole (SMZ), Tetracyclines: TET, and Diaminopyridines: Trimethoprim (TMP)) were captured and then utilized as feature values for the identification of classes using machine learning.

In another study, a synergistic gold/silver bimetallic NC was used as a probe for the simultaneous detection of TET, AMP, and sulfacetamide [121]. Using the multivariate curve resolution-alternating least-squares method, good linear ranges of 5–5000 and 50–5000 ng/mL were achieved for AMP detection and TET and sulfacetamide detection, respectively.

### 4.4. Foodborne Pathogens

Foodborne pathogens can contaminate the food we consume, threatening human health and causing economic losses. They can lead to food poisoning, endocarditis, and sepsis [122]. Over the past few decades, nanomaterial-based fluorescent biosensors have been widely used to monitor foodborne pathogens.

*S. aureus* and *E. coli* are typical foodborne pathogens, so numerous efforts have been made to detect them. For example, Huang et al. prepared core–shell structural fluorescent magnetic nanobeads (FMNBs) to perform a lateral flow immunoassay for *E. coli* [123]. Owing to the magnetic and fluorescent properties of FMNBs, this immunoassay allowed the quantified detection of 2.39 × 10^2^ CFU/mL for *E. coli*. Similarly, Zhao et al. prepared an immunosensor based on microspheres labeled with CDs to detect *E. coli* in milk [124]. CD-microspheres were prepared using *S. aureus* cells as carriers to incorporate the CD particles. In combination with the immunomagnetic bead technique, a CD-microsphere immunosensor was successfully used for *E. coli* detection.

Moreover, other nanomaterials can be used to construct fluorescent biosensors to detect foodborne pathogens. Liu et al. designed a fluorescent genosensor for *S. aureus* using graphene oxide quantum dots (GOQDs) and ssDNA-modified UCNPs [125]. The ssDNA was first conjugated with the UCNPs, followed by adsorption onto the surface of the GOQDs via π-π stacking and hydrogen bonding, which quenched the fluorescence of the UCNPs. However, the presence of the complementary nuc target triggered the release of the GOQDs, leading to fluorescence recovery of the UCNPs. A good linear range of 1 × 10^−17^ to 1 × 10^−11^ M was observed for *S. aureus* detection. To improve separation efficiency, Li et al. prepared aptamer-modified fluorescent magnetic nanoprobes (apt-FMNPs) to simultaneously detect multiple pathogens [126]. As fluorescence probes, the FMNPs were composed of magnetic γ-Fe_2_O_3_ and fluorescent QDs. Then, aptamers of *E. coli* and *Salmonella typhimurium* were separately conjugated with different FMNPs to obtain apt-FMNP nanoprobes. According to the different magnetic responses of the pathogen–nanoprobe composites under external magnetic fields, the target bacteria were removed via magnetic separation and subjected to fluorescence recording. Thus, multiple pathogens were accurately detected using apt-FMNPs. This apt-FMNP-based strategy for the simultaneous detection of multiple pathogens is promising for the analysis of foodborne pathogens.

### 4.5. Other Illegal Additives

Agro-products often contain multiple contaminants, such as pesticides, environmental hormones, and illegal food additives. These residues can enter the human body and cause serious damage to human health. Therefore, the development of rapid and sensitive methods for contaminant detection in agro-products is urgently required.

OPs are often used because of their insecticidal toxicity, which leads to their coexistence in the food matrix. Zhang et al. designed a multi-analyte fluorescence immunoassay for the simultaneous detection of three OPs (triazophos, parathion, and chlorpyrifos) using an AuNP-assisted signal amplification strategy [127]. Detection probes based on AuNPs were prepared by simultaneously modifying AuNPs with antibodies and fluorescently labeled oligonucleotides. The developed immunoassay achieved linear detection ranges of 0.01–20, 0.05–50, and 0.5–1000 μg/L for triazophos, parathion, and chlorpyrifos, respectively. Wang et al. proposed a multiplexed biosensor based on FRET from multicolor UCNPs to single black phosphorus nanosheets (BPNSs) to simultaneously detect paraquat and carbendazim pesticides [128]. The aptamer-modified UCNPs were adsorbed onto BPNS via π–π stacking, which quenched the fluorescence of the UCNPs. After adding paraquat and carbendazim, the aptamers preferentially bonded with their corresponding targets, leading to fluorescence recovery. The developed aptasensor provides insight into the simultaneous monitoring of multiple targets.

Residual environmental hormones are becoming a public concern because of their severe risks to human health. Amalraj et al. designed a fluorescent biosensor for the simultaneous detection of CAP and 17β-estradiol based on a MOF-MoS_2_ nanomaterial and an enzyme-assisted signal amplification strategy [129]. The MOF coupled with MoS_2_ sheets (MOF-MoS_2_) was used as an efficient fluorescent nanoquencher. The aptamers for CAP and 17β-estradiol were separately adsorbed onto MOF-MoS_2_, leading to decreased fluorescence owing to FRET between the fluorophores and MOF-MoS_2_. Upon the addition of CAP and 17β-estradiol, the corresponding aptamers were preferentially bound to the targets, followed by partial recovery of the fluorescence. The addition of Exo I led to the digestion of the aptamer, resulting in remarkable fluorescence recovery. Simultaneously, the released targets triggered the next cycle of signal amplification. Under optimal conditions, the sensor achieved a LOD of 180 and 200 pM for 17β-estradiol and CAP, respectively.

Nowadays an increasing number of works about nanomaterial-based fluorescent biosensors have been reported for food safety analysis. A comparison of different fluorescent biosensors in food safety analysis is listed in Table 1. 

## 5. Conclusions and Perspectives

The detection of harmful substances in food is crucial for ensuring food safety. The current detection methods mainly depend on expensive instruments. Although these methods can detect various targets with high performance, they require complex sample pretreatment and professional staff and incur high costs. As an alternative, nanomaterial-based fluorescent biosensors can be used to address these issues regarding food safety.

The key to developing fluorescent biosensors is preparing fluorescent materials with excellent biocompatibility, high efficiency, and good stability. Herein, we summarize the commonly used biomaterials and their applications in fluorescent biosensors. Despite this progress, there are gaps that must be addressed. The potential directions for future research are presented below.

(1)With the development of bio- and nanomaterials, the development of inexpensive, easy-to-synthesize, and eco-friendly materials remains a topic for future research.(2)Many nanomaterial-based biosensors have been successfully developed; however, their detection performance must be improved. Some suffer from limitations, such as low stability, poor repeatability, and a weak anti-interference ability. Therefore, there is an urgent need to develop efficient methods for fluorescence biosensing.(3)High-performance aptamers have been screened. Aptamer screening has accelerated the development of new aptasensors. Although numerous aptasensors have been developed, not all types of analyte have been investigated. Biomaterial analogs should be developed and integrated into biosensors to increase the number of aptamers.(4)Currently, newly developed biosensors are still in their early stages. Future research in this area should focus on real-time monitoring or on-site analysis, for example, the investigation of portable devices for food safety analysis.(5)Although nano- and biomaterials have advantages in terms of performance, their synthesis conditions must be optimized. In addition to full purification and the removal of impurities, multiple fluorescences should be used instead of a single fluorescence.(6)Increased practical applications. Currently, most fluorescent systems are in the experimental stage. The practical applications of nanomaterials-based fluorescent biosensors in complex matrices remain a great challenge. Adopting machine learning and microfluidic systems into fluorescence biosensors may meet the criteria of cheap real-time detection in complex matrices.

## Figures and Tables

**Figure 1 biosensors-12-01072-f001:**
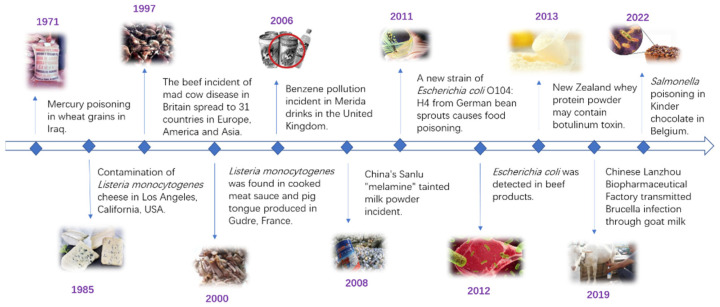
A brief timeline of the typical food safety issues.

**Figure 2 biosensors-12-01072-f002:**
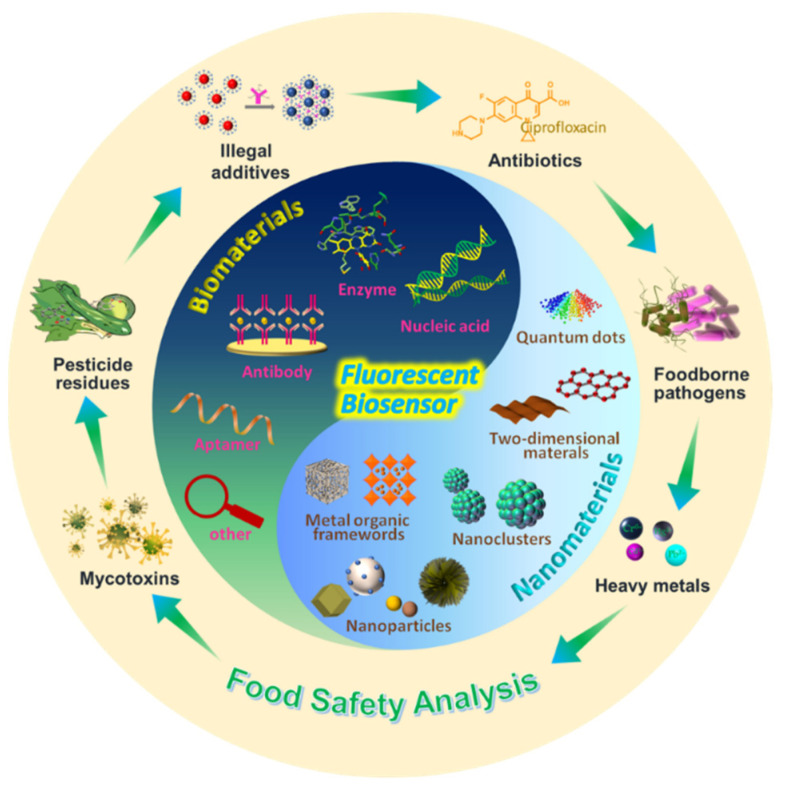
Schematic illustration of nanomaterial-based fluorescent biosensors for food safety analysis.

**Figure 3 biosensors-12-01072-f003:**
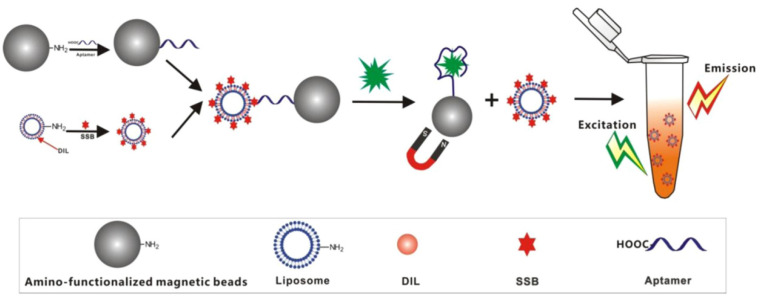
Scheme of a fluorescent biosensor for the CAP assay using a combination of MBs and fluorescent liposomes. Reprinted from [48] with permission from Elsevier.

**Figure 4 biosensors-12-01072-f004:**
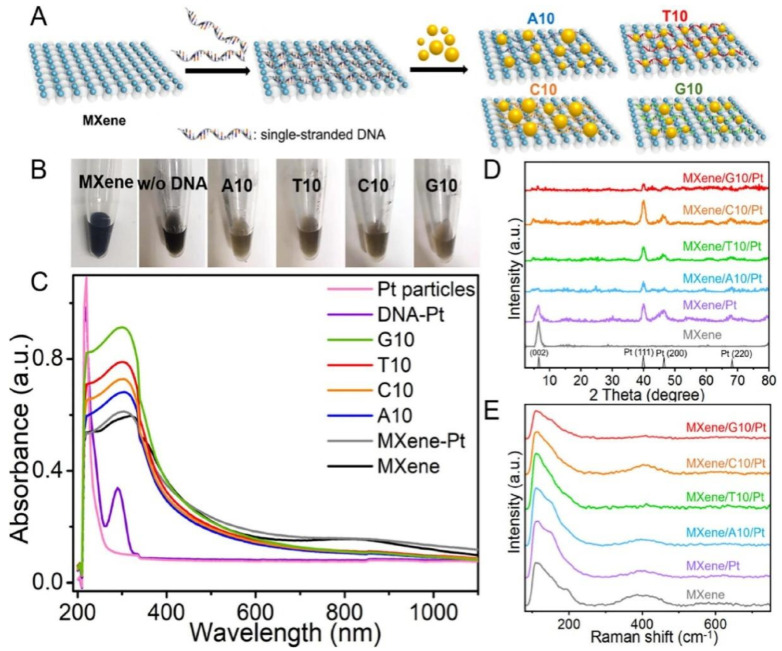
DNA-mediated synthesis of PtNPs on MXene nanosheets for the colorimetric Hg^2+^ assay. (**A**) Scheme of the DNA-mediated synthesis of PtNPs on MXene nanosheets. (**B**) The color of DNA-encoded MXene Nanosheet-Pt solution synthesized with distinct DNA sequences. (**C**) UV/vis absorption spectra, X-ray diffraction (XRD) patterns (**D**), and Raman spectra (**E**) of different materials as indicated. Reprinted from [68] with permission from Elsevier.

**Figure 5 biosensors-12-01072-f005:**
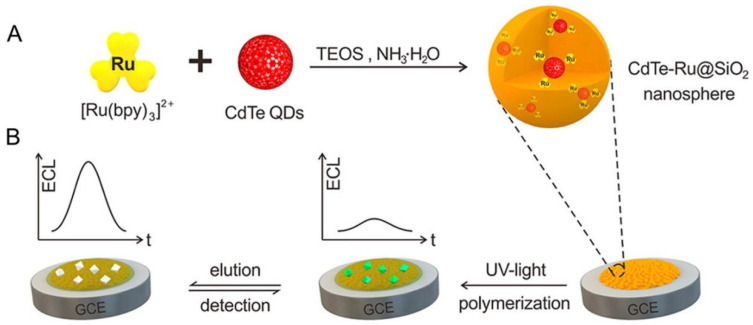
Schematic illustration of the synthesis of CdTeRu@SiO_2_ nanospheres (**A**) and fabrication of the MIP-ECL platform for mycotoxin detection (**B**). Reprinted from [75] with permission from the American Chemical Society.

**Figure 6 biosensors-12-01072-f006:**
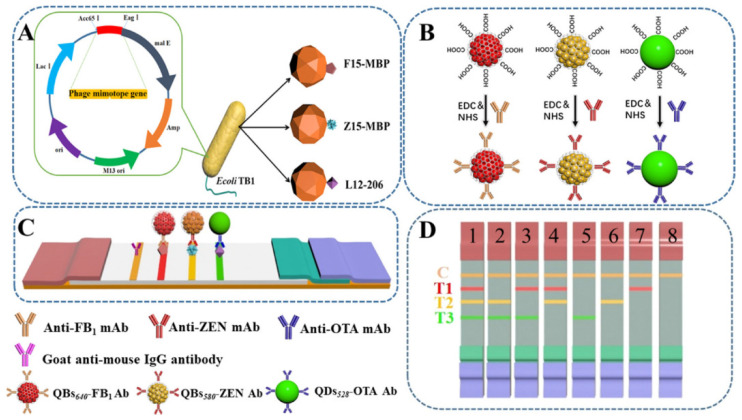
Scheme of the biosynthetic mycotoxin conjugate mimetics for multiplex mycotoxin immunochromatographic assays. (**A**) Biological expression strategy of peptide-MBP fusion protein; (**B**) fabrication process of the prepared QDs/QBs-mAb probes; (**C**) schematic illustration of the tricolor mICA; (**D**) schematic illustration for the interpretation of test results. (1) negative; (2) FB1; (3) ZEN; (4) OTA; (5) FB1 and ZEN; (6) FB1 and OTA; (7) ZEN and OTA; and (8) FB1, ZEN, and OTA. Reproduced with permission from [90]. Copyright (2020) American Chemical Society.

**Figure 7 biosensors-12-01072-f007:**
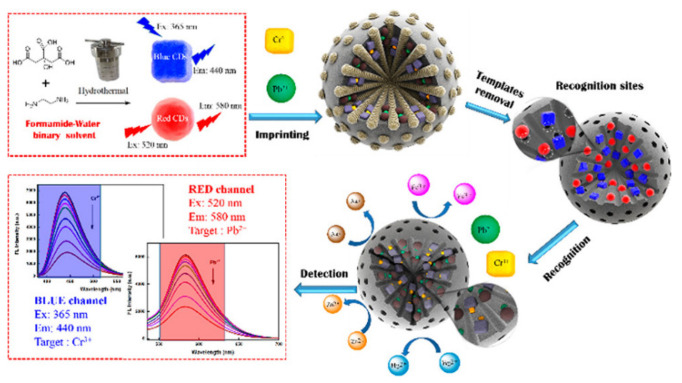
Schematic illustration of the preparation process and simultaneous detection of Cd^2+^ and Cu^2+^ using CD-incorporated ion-imprinted fluorescent polymers. Reproduced with permission from [101]. Copyright (2019) American Chemical Society.

**Table 1 biosensors-12-01072-t001:** Reported Nanomaterials-based fluorescent biosensor for the determination of food contaminants.

Analytes	Nanomaterials	Linear Range	LOD	Food Matrix	Ref.
**Mycotoxins**					
T-2 toxin	AgNCs	0.005–500 ng/mL	0.93 pg/mL	maize and wheat	[86]
DON	QDs	1–10 ng/ml	28 μg/kg	wheat	[87]
FB_1_	AuNPs	7.3–22.6 ng/mL	1.1 ng/mL	wheat	[89]
FB_1_, ZEN and OTA	QDs/QD	------	0.25 ng/mL 3.0 ng/mL 0.5 ng/mL	------	[90]
OTA and AFB_1_	hydrogel particles	0.1–500 ng/mL, 0.1–200 ng/mL	0.1 ng/mL	corn flour	[91]
OTA, AFB_1_, and FB_1_	TiO_2_-Si	0.1–10 ng/mL, 0.01–10 ng/mL, 0.001–10 ng/mL	15.4 pg/mL 1.48 pg/mL 0.21 pg/mL	rice, corn, and wheat	[92]
**Heavy metals**					
Hg^2+^	CdTe QDs	0.7–900 nM	2.5 nM	water	[94]
Hg^2+^	DNA-hydrogel	------	10 nM	water	[95]
Pb^2+^	NH2-MIL-125(Ti) MOF	0–11 nM	7.7 pM	------	[96]
Pb^2+^	AuNCs	0–190 nM	10 nM	pond water and river water	[97]
Pb^2+^	ZIF-8	0.01–10.0 nM	7.1 pM	water and fish	[98]
Cd^2+^ and Pb^2+^	V_6_O_13_ nanobelts	5–200 μg/L, 5–100 μg/L	1.89 μg/L, 2.11 μg/L	water	[100]
Cr^3+^ and Pb^2+^	CDs	0.1–6.0 μM, 0.1–5.0 μM	27 nM 34 nM	water	[101]
Cd^2+^, Hg^2+^ and Pb^2+^	DNAzymes	------	4.8 nM 2.0 nM 0.1 nM	------	[102]
Hg^2+^ and Ag^+^	MPDA frame-works	0–2 nM 1–3 nM	1.3 pM 34 pM	water	[103]
**Antibiotics**					
TET	Apt-tet MBs	0.001–10 ng/mL	0.724 pg/mL	fish and honey	[108]
TET OTC DOX	BNQD-Eu^3+^	------	0.019 μM 0.104 μM 0.104 μM	milk and beef	[109]
TET	BSA-BMNCs	------	------	water and milk	[110]
KAN	aptamer probes	0.1–75 nM	0.097 nM	aquatic products	[112]
tobramycin	magnetic beads	0.3–50 μM	17.37 nM	------	[113]
AMP	UCNPs	10–100 ng/mL	3.9 ng/mL	milk	[115]
penicillin	CDs	1–32 nM	0.34 nM	milk	[116]
CAP	DNA four-arm junctions	1.0 pg/mL–10 ng/mL	0.72 pg/mL	milk and honey	[119]
AMP, CPFX, KAN, SMZ, TET, and TMP	carbon nanoparticle	------	------	poultry feeds	[120]
TET, AMP, and sulfacetamide	nanocluster	50–5000 ng/mL 5–5000 ng/mL 50–5000 ng/mL	3.5 ng/mL 1.4 ng/mL 7.6 ng/mL	milk	[121]
**Foodborne pathogens**					
*Escherichia coli*	CDs	2.4 × 10^2^–2.4 × 10^7^ CFU/mL	2.4 × 10^2^ CFU/mL	milk	[124]
*S. aureus*	GOQDs	1 × 10^−17^–1 × 10^−11^ mol/L	0.98 × 10^−17^ mol/L	------	[125]
*Escherichia coli, Salmonella typhimurium*	apt-FMNPs	40–10^8^ CFU/mL 63–10^8^ CFU/mL	16 CFU/mL 25 CFU/mL	------	[126]
**Other illegal additives**					
Triazophos, parathion, chlorpyrifos	AuNPs	0.01–20 μg/L, 0.05–50 μg/L, 0.5–1000 μg/L	0.007 μg/L, 0.009 μg/L, 0.087 μg/L	rice, wheat, cucumber, cabbage, and apple	[127]
paraquat, carbendazim pesticides	UCNPs and BPNS	1.0–1.0 × 10^5^ ng/mL	0.18 ng/mL, 0.45 ng/mL	------	[128]
CAP, 17β-estradiol	MOF-MoS_2_	0.9917–5 nM, 0–5 nM	200 pM, 180 pM	milk, honey, and water	[129]

## Data Availability

Not applicable.

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
