# Peer review of "Nanomaterial-Based Fluorescent Biosensor for Food Safety Analysis"

_biosensors, 2022, doi:10.3390/bios12121072_

Round 1

Reviewer 1 Report

The manuscript " Nanomaterial-based fluorescent biosensor for food safety analysis" was submitted to Biosensors, which describes biomaterials and nanomaterials to establish fluorescent biosensors.

From my point of view, there are no serious concerns or obstacles in this review that need to be further clarified and revised by the authors. According to my opinion, this manuscript is logically organized, well referenced, and represents a valuable contribution. However, minor changes have to take into account:.

Q1. The first reference in the introduction is too old and should refer to the latest food industry standards.

Q2. Please correct the grammar errors throughout the manuscript.

Q3. The Table 1 is not included properly in the text. The authors have to indicate in the text that Table 1 summarizes the point 4

Author Response

Responses to the Reviewers’ Comments and Corresponding Revisions

Responses to the comments of Reviewer 1:

General Comments:

The manuscript " Nanomaterial-based fluorescent biosensor for food safety analysis" was submitted to Biosensors, which describes biomaterials and nanomaterials to establish fluorescent biosensors.

From my point of view, there are no serious concerns or obstacles in this review that need to be further clarified and revised by the authors. According to my opinion, this manuscript is logically organized, well referenced, and represents a valuable contribution. However, minor changes have to take into account:

Response: Thanks sincerely for the reviewer who gave valuable suggestions to improve our manuscript and thank you for your appreciation of our work. We have revised the manuscript according to your suggestions as follows.

Comment 1: The first reference in the introduction is too old and should refer to the latest food industry standards.

Response: Thank you for your valuable advice. we made the corresponding modifications in the revised manuscript as follows:

“References:

  1. Dzwolak, W. Assessment of HACCP plans in standardized food safety management systems – The case of small-sized Polish food businesses. Food Control. 2019, 106, 106716.”

Comment 2: Please correct the grammar errors throughout the manuscript.

Response: Thank you very much for the comments and valuable suggestions to improve our manuscript. We have asked a professional editing service to check English of our manuscript. And accordingly, we have carefully revised our expressions throughout the manuscript.

Comment 3: The Table 1 is not included properly in the text. The authors have to indicate in the text that Table 1 summarizes the point 4.

Response: Thank you very much for the comments and valuable suggestions to improve our manuscript. Accordingly, we revised our manuscript as follows:

“Fluorescent biosensors have been widely applied in food safety analysis due to their distinct characteristics in selective capture and sensitive analysis of analytes. A comparison of different fluorescent biosensors in food safety analysis is listed in Table 1.”

Reviewer 2 Report

1) The article is well written, focuses on all areas considered, but largely reflects work done in Asia/China, consider changing the title to:

Nanomaterial-based fluorescent biosensor for food safety analysis in Asia,

or

Nanomaterial-based fluorescent biosensor for food safety analysis in China.

or alternatively, to avoid changing the title of the article, place more references from other authors working in the same area from other regions of the world. In this case, you could improve or increase the number of references in table 1.

2) avoid cutting the word "analysis" in the title.

3) improve the design of figure 1: move the text "Biomaterials" and "Nanomaterials" so it doesn't overlap over the arrows.

4) Your article uses the word "attracted" 13 times, try to use another word or change the text so you don't repeat the same word so much.

Author Response

Responses to the Reviewers’ Comments and Corresponding Revisions

Responses to the comments of Reviewer 2:

Comment 1: The article is well written, focuses on all areas considered, but largely reflects work done in Asia/China, consider changing the title to:

Nanomaterial-based fluorescent biosensor for food safety analysis in Asia,

or

Nanomaterial-based fluorescent biosensor for food safety analysis in China.

or alternatively, to avoid changing the title of the article, place more references from other authors working in the same area from other regions of the world. In this case, you could improve or increase the number of references in table 1.

Response: Thank you very much for the constructive comments to improve our manuscript. Accordingly, we have cited more references from other authors working in the same area from other regions of the world. Accordingly, we revised the number of references in Table 1.

Comment 2: avoid cutting the word "analysis" in the title.

Response: We thank the reviewer for the comments and valuable suggestions to improve our manuscript. The manuscript is still titled with “Nanomaterial-based fluorescent biosensor for food safety analysis”.

Comment 3: improve the design of figure 1: move the text "Biomaterials" and "Nanomaterials" so it doesn't overlap over the arrows.

Response: Thank you very much for the valuable advice. We made the corresponding revisions as follows:

Comment 4: Your article uses the word "attracted" 13 times, try to use another word or change the text so you don't repeat the same word so much.

Response: Thank you very much for the comments and valuable suggestions to improve our manuscript. We have asked a professional editing service to check English of our manuscript. And accordingly, we have carefully revised our expressions throughout the manuscript. For example, we have replaced “attracted” with “gathered”, “drawn”, “received”, “grasped”, etc.

Thank you again for your valuable comments.

Reviewer 3 Report

The manuscript entitled: “Nanomaterial-based fluorescent biosensor for food safety analysis”, reference: biosensors-2021027

In my understanding the manuscript is focused in a highly important subject, nevertheless it has several important issues, that I hope that the authors will address in an extensive and rigorous way, since it will require a considerable effort.

General comments:

For a review, in my opinion, the Introduction describes very little examples for such an important subject. It should include several examples, and statistics, such as how is the food safety evolving? How many people require hospital assistance due to food poisoning of this, this and that compound? I understand that is impossible to include all examples, and, in a matter of fact, it would be useful if the authors defined a reasonable time period for all the examples that should be included.  All this would promote a bigger picture, the trends and support the statement that this is indeed a highly relevant subject. Moreover, the authors should approach the issue of food waste and spoilage, another highly relevant subject that is intrinsically related to the main focus of this manuscript.

In my point of view, all the approaches are very superficial, in particular, sections 2.1., 2.2, 2.3, 2.4. In addition, I found hard to distinguish 2.5 from 2.1.

It is hard to grasp the required technologies/equipments during the assessment of the descried fluorescent biosensors, thus it is hard to understand if these approaches are feasible in the food industry or cost effective. Therefore, I urge the authors to clearly state the advantages and the disadvantages or foreseen Achilles heel of each florescent biosensor.

Poin-by-point comments

Figure 1 is, in my understanding a bit confusing, as it is over simplified. Can the authors please try to improve its quality and clarity.

Line 260, I failed to find the acronym for MOFs

All species names must be written in full on their first appearance in the manuscript.

Table 1, please revise or clarify: "S. typ" and "river"

The manuscript is easy to read, however it is over simplified, and several personal pronouns are used, which in my opinion should be avoided in scientific writing.

Author Response

Responses to the Reviewers’ Comments and Corresponding Revisions

Responses to the comments of Reviewer 3:

The manuscript entitled: “Nanomaterial-based fluorescent biosensor for food safety analysis”, reference: biosensors-2021027.

In my understanding the manuscript is focused in a highly important subject, nevertheless it has several important issues, that I hope that the authors will address in an extensive and rigorous way, since it will require a considerable effort.

General comments:

For a review, in my opinion, the Introduction describes very little examples for such an important subject. It should include several examples, and statistics, such as how is the food safety evolving? How many people require hospital assistance due to food poisoning of this, this and that compound? I understand that is impossible to include all examples, and, in a matter of fact, it would be useful if the authors defined a reasonable time period for all the examples that should be included. All this would promote a bigger picture, the trends and support the statement that this is indeed a highly relevant subject. Moreover, the authors should approach the issue of food waste and spoilage, another highly relevant subject that is intrinsically related to the main focus of this manuscript.

Response: Thanks sincerely for the reviewer who gave valuable suggestions to improve our manuscript and thank you for your appreciation of our work. We have carefully revised the manuscript according to the comments. A literature survey using Web of Science shows that more than 100,000 journal articles have been published about food safety after 2011. A brief timeline of the typical food safety issues is shown in Figure 1. Currently, the most significant food safety issues are caused by the consumption of contaminated food. Food contamination may derive from a variety of sources, such as food waste and spoilage, the abuse of pesticides, foodborne pathogenic bacteria contamination, the production of toxins, and the formation of harmful chemicals during food processing. Among these, chemical contamination has obtained increased attention because the chemical contaminants such as toxins, heavy metals, pesticides, and antibiotic residues can hardly be degraded and easily accumulate in the human body, which may cause severe harm to human health. Hence, food safety analysis is of great importance.

Figure 1 A brief timeline of the typical food safety issues.

In my point of view, all the approaches are very superficial, in particular, sections 2.1., 2.2, 2.3, 2.4. In addition, I found hard to distinguish 2.5 from 2.1.

Response: Thank you very much for the constructive comments and valuable suggestions to improve our manuscript. We have rewritten the introduction part in the revised manuscript. We introduced the biomaterials in food safety analysis. For example, aptamers and antibodies are used to design aptasensors and immunosensors, respectively. In sections 2.5, we demonstrated that the single stranded DNA binding protein uses a unique folding pattern to specifically bind to single stranded DNA. This concept was employed by Miao et al. in 2016 (Ref. 48). This kind of protein is different from antibodies (2.2. Antibodies) and enzymes (2.3. Enzymes). To facilitate understanding, we introduced an example (Ref. 48).

Thank you again for the valuable comment.

Ref. 48: Miao, Y.-B.; Ren, H.-X.; Gan, N.; Cao, Y.; Li, T.; Chen, Y.; Fluorescent aptasensor for chloramphenicol de-tection using DIL-encapsulated liposome as nanotracer. Biosens. Bioelectron. 2016, 81, 454-459.

It is hard to grasp the required technologies/equipments during the assessment of the descried fluorescent biosensors, thus it is hard to understand if these approaches are feasible in the food industry or cost effective. Therefore, I urge the authors to clearly state the advantages and the disadvantages or foreseen Achilles heel of each florescent biosensor.

Response: We thank the reviewer for the valuable comments and suggestions to improve our manuscript. As the reviewer suggested, we did carefully revisions of our manuscript for clarity. The advantages and the disadvantages have been added in the revised manuscript. For example, we made the revisions as follows:

“Peltomaa et al. proposed a homogeneous fluorescence immunoassay based on the flu-orescence quenching ability of AuNPs and a recombinant epitope-mimicking fusion protein for the detection of mycotoxin fumonisin B1 (FB1) [89]. The fumonisin mimotope was used as a fluorescent protein for FB1 detection without the need of a secondary antibody. This immunoassay could be carried out in a single step without cumbersome washing steps. There was no significant cross-reactivity with other mycotox-ins and acceptable recoveries were obtained from spiked wheat samples, indicating a great promise for simple analysis of mycotoxin-contaminated food samples.”

In the discussion part, we discussed the parameters of fluorescent biosensors. On the one hand, most of the existing biosensors for food safety analysis could provide satisfactory sensitivities. However, few of them are widely used in practical applications due to the cost, usability, and speed of analysis. The goals of food analysis, including fluorescent biosensors and other methods, should be focused on the selectivity, reproducibility, and stability in complex matrixes and miniaturization of biosensors by technologies. Adopting machine learning and microfluidic systems into fluorescent biosensor may meet the criteria of cheap real-time detection in complex matrices.

From my point of view, there are no serious concerns or obstacles in this review that need to be further clarified and revised by the authors. According to my opinion, this manuscript is logically organized, well referenced, and represents a valuable contribution. However, minor changes have to take into account:

Response: Thanks sincerely for the reviewer who gave valuable suggestions to improve our manuscript and thank you for your appreciation of our work. We have carefully revised the manuscript according to the comments.

Comment 1: Figure 1 is, in my understanding a bit confusing, as it is over simplified. Can the authors please try to improve its quality and clarity.

Response: Thank you for your valuable advice. We made the corresponding revisions as follows:

Figure 2. Schematic illustration of nanomaterial-based fluorescent biosensors for food safety analysis.

Comment 2: Line 260, I failed to find the acronym for MOFs.

Response: Thank you very much for the comments and valuable suggestions. According to these suggestions, we made the corresponding modifications in the revised manuscript as follows:

“3.4. Metal-organic frameworks

Metal-organic frameworks (MOFs) are built from metal ions and organic linkers [71].”

Comment 3: All species names must be written in full on their first appearance in the manuscript.

Response: Thank you very much for the comments and valuable suggestions to improve our manuscript. And accordingly, we have carefully revised our expressions throughout the manuscript.

Comment 4: Table 1, please revise or clarify: "S. typ" and "river".

Response: Thank you very much for the comments and valuable suggestions to improve our manuscript. Accordingly, we have revised “S. typ to” to “Salmonella typhimurium”. "river" has been deleted.

Comment 5: The manuscript is easy to read, however it is over simplified, and several personal pronouns are used, which in my opinion should be avoided in scientific writing.

Response: Thank you very much for the comments and valuable suggestions to improve our manuscript. We have asked a professional editing service to check English of our manuscript. And accordingly, we have carefully revised our expressions throughout the manuscript.

Round 2

Reviewer 3 Report

The manuscript entitled: “Nanomaterial-based fluorescent biosensor for food safety analysis”, biosensors-2021027

I must congratulate the authors for the enormous improvement of the manuscript figures. They are now clearer and informative.

As for the information added to the manuscript text, the authors performed improvements in terms of clarity, quality and scientific soundness.

Line 429, ssDNA acronym should be defined at its first appearance, and systematically used throughout the manuscript. Please thoroughly revise the manuscript for uniformity of the acronyms.

Author Response

Responses to the comments of Reviewer 3:

General comments:

The manuscript entitled: “Nanomaterial-based fluorescent biosensor for food safety analysis”, biosensors-2021027

I must congratulate the authors for the enormous improvement of the manuscript figures. They are now clearer and informative.

As for the information added to the manuscript text, the authors performed improvements in terms of clarity, quality, and scientific soundness.

Line 429, ssDNA acronym should be defined at its first appearance, and systematically used throughout the manuscript. Please thoroughly revise the manuscript for uniformity of the acronyms.

Response: Thanks sincerely for the reviewer who gave valuable suggestions to improve our manuscript and thank you for your appreciation of our work. We have carefully revised the manuscript according to the comments. For example, ssDNA acronym is defined at Line 167. Accordingly, we have checked all the acronym in the revised manuscript.
